# Clocking mechanism from a minimal spinning particle model

Tobiasz Pietrzak[1*] and Łukasz Bratek[2]

**1** H. Niewodniczański Institute of Nuclear Physics, Polish Academy of Sciences,
ul. Eljasza-Radzikowskiego 152, PL 31342 Kraków, Poland
**2** Institute of Physics, Cracow University of Technology,
ul. Podchorażych 1, PL-30084 Kraków, Poland

⋆ tobiasz.pietrzak@ifj.edu.pl

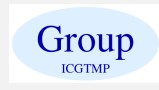
## Abstract

The clock hypothesis plays an important role in the theory of relativity. To test this hypothesis, a mechanical model of an ideal clock is needed. Such a model should have the phase of its intrinsic periodic motion increasing linearly with the affine parameter of the clock's center of mass worldline. A class of relativistic rotators introduced by Staruszkiewicz in the context of an ideal clock is studied. A singularity in the inverse Legendre transform leading from the Hamiltonian to the Lagrangian leads to new possible Lagrangians characterized by fixed values of mass and spin. In free motion the rotators exhibit intrinsic motion with the speed of light and fixed frequency.

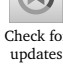

## 1 Introduction

By definition an ideal clock always measures its proper time. The equality of time measured by natural clocks and that of ideal clock has been verified to a high degree of precision [1], however it is not known whether this equality always holds true. Discrepancies could occur for extreme accelerations of order $c^2/L$ where $L$ is a length scale characterising a given system (e.g. $10^{29}\frac{m}{s^2}$ for electron's Zitterbewegung frequency). Accelerations that high are not yet experimentally attainable. Nevertheless, an attempt can be made to theoretically test the clock hypothesis (which refers to classical concepts) within the same framework one uses to describe real mechanical systems. In this respect a classical model of the ideal clock must be devised.[1]

---

[1]A spatially extended quantum field-theoretical model of a clock devised in the clock hypothesis context [2] goes beyond this conceptual limitation. The authors concluded that *no device built according to the rules of quantum field theory can measure proper time along its path*. It is also known that for any timelike worldline in any spacetime, there is a sufficiently small light clock that accurately measures the proper time [3], however this kind of clock is not a mechanical system.



As a purely mathematical construct unrelated to any material mechanism, an ideal clock would be a simple non-quantum device. The mechanism of such a clock could be designed in the following way. In the momentum rest frame, the image of the spatial direction of the Pauli-Lubański four-vector could be identified with the equator on the Riemann sphere of null directions and used as the clock's face. On the other hand, the image of a null direction (carrying the spinning degrees of freedom) would be a point moving about the equator, counting the number of times the phase has been increased by $2\pi$, and thus represent the clock's hand.

Such a model has been proposed by Staruszkiewicz [4]. It is based on the concept of a relativistic rotator – a dynamical system described by position, single null direction (thus with 5 degrees of freedom) and, additionally, two-dimensional parameters – mass $m$ and length $l$ used to set the values of Casimir invariants, respectively, to $m^2$ and $-\frac{1}{4}m^4l^2$. It seems that the model provides the simplest mechanical system whose clocking frequency could be fixed this way. Among the entirety of Lagrangians possible for the family of relativistic rotators considered in [4], there are only two which satisfy the last requirement above. As later shown at the Lagrangian level [5] the unique Lagrangians are defective when interpreted as dynamical systems with 5 degrees of freedom (the Hessian rank is 4, not 5). This explained the observation [6] that in free motion of the clock the phase, and hence the clocking frequency, remained indeterminate as functions of the proper time of the center of momentum frame, contrary to the original motivation.

A possible way to find the required Lagrangian and stabilise the clocking frequency leads through the inverse Legendre transformation (from the Hamiltonian to a Lagrangian). As observed for the rotator in [7], a singularity in this transformation distinguishes intrinsic motion with the speed of light. This changes the analytic form of the required Lagrangian.[2]

## 2 Staruszkiewicz class of relativistic rotators.

A class of relativistic rotators is defined by the following Hamiltonian action introduced by Staruszkiewicz [4]

$$S = -m \int d\lambda \sqrt{\dot{x}\dot{x}} f(\xi), \qquad \xi \equiv -l^2 \frac{\dot{k}\dot{k}}{(k\dot{x})^2}, \quad f'(\xi) \not\equiv 0. \qquad (1)$$

Here,[3] the dot denotes differentiation with respect to $\lambda$ – an arbitrary parameter along the worldline, and $f$ can be arbitrary non-constant and positive function of a reparametrization invariant argument $\xi$ depending on the spinning degrees of freedom through a null direction $k$ (the latter property means that $\xi$ must be a Poincaré scalar, independent of arbitrary scale of null vector $k$).

Representations of the Poincaré group are enumerated by the eigenvalues of two Casimir operators (for the case of massive representations). These operators are the square of the momentum four-vector $C_1 = p^\mu p_\mu$ and the square of the Pauli-Lubański four-vector $C_2 = W^\mu W_\mu$, where:

$$W^\mu = \frac{1}{2}\varepsilon^{\mu\nu\alpha\beta}p_\nu M_{\alpha\beta}, \qquad M_{\alpha\beta} = x_\alpha p_\beta - x_\beta p_\alpha + \Sigma_{\alpha\beta}.$$

The expression $\Sigma_{\alpha\beta}$ represents the internal angular momentum (spin). To find suitable Lagrangians in the considered class of rotators one can proceed as follows. The conserved quantities $p_\alpha$ and $M_{\alpha\beta}$ are determined from the action (1) (with $\Sigma_{\alpha\beta} = k_\alpha \pi_\beta - k_\beta \pi_\alpha$), where the

---

[2]These results can be considered new as they are based on yet unpublished paper [7].

[3]Throughout this paper $x^\mu$ denotes the position vector, $k^\mu$ is the single null direction carrying the spinning degrees of freedom. The scalar product is denoted by $xy \equiv \eta_{\alpha\beta}x^\alpha y^\beta = x^\alpha y_\alpha$ (Einstein's summation convention is used), where $(\eta_{\alpha\beta}) = \text{diag}(1, -1, -1, -1)$, and $\epsilon^{0123} = 1$ for the Levi-Civita completely anti-symmetric pseudo-tensor. Greek indices run over $0, 1, 2, 3$ and $0$ stands for the time component.

momenta canonically conjugated to $x^\mu$ and $k^\mu$ read, respectively,

$$p_\mu = m\left[f(\xi)\frac{\dot{x}_\mu}{\sqrt{\dot{x}\dot{x}}} - 2\xi f'(\xi)\frac{\sqrt{\dot{x}\dot{x}}}{k\dot{x}}k_\mu\right], \qquad \text{and} \qquad \pi_\mu = 2m\frac{\sqrt{\dot{x}\dot{x}}}{k\dot{k}}\xi f'(\xi)\dot{k}_\mu.$$

The corresponding Casimir invariants can now be calculated

$$C_1 = m^2[f^2(\xi) - 4\xi f(\xi)f'(\xi)], \qquad C_2 = -4m^4 l^2 \xi f^2(\xi)[f'(\xi)]^2.$$

By requiring that $C_1 \equiv m^2$ and $C_2 \equiv -\frac{1}{4}m^4 l^2$ (identically), one gets two first-order differential equations that, remarkably, have a common solution of the form $f(\xi) = \sqrt{1 \pm \sqrt{\xi}}$. The Hamiltonian action describing these rotators takes on the form

$$S = -m\int d\lambda \sqrt{\dot{x}\dot{x}}\sqrt{1 \pm \sqrt{-l^2\frac{\dot{k}\dot{k}}{(k\dot{x})^2}}} + \int d\lambda\, \Lambda kk, \tag{2}$$

with $\Lambda$ being a Lagrange multiplier. As will be explained below, the dynamical system defined by the action (2) is not suitable as a clock. However, it is equivalent to a geometric model of a spinning particle introduced earlier in a different context by Lyakhovich, Segal, and Sharapov [8] and as such can be used with success.

## 3   Hessian rank deficiency for subluminal intrinsic motion

In the Lagrangian form of dynamics, there are $s$ Lagrangian equations

$$\frac{d}{d\lambda}\frac{\partial L}{\partial \dot{q}^i} - \frac{\partial L}{\partial q^i} = 0, \qquad i = 1, 2, \ldots, s,$$

for a dynamical system with $s$ (physical) degrees of freedom. In this form the Lagrangian $L$ is assumed to be a function of $s$ generalised coordinates $q^i = Q^i(\lambda)$ and the corresponding velocities $v^i = \dot{Q}^i(\lambda)$ that altogether characterise the physical state of the system. Differentiating the Lagrange equations with respect to the independent parameterization $\lambda$, one gets a system of second-order equations

$$H_{ij}a^j = \frac{\partial L}{\partial q^i} - \frac{\partial^2 L}{\partial v^i \partial q^j}v^j - \frac{\partial^2 L}{\partial \lambda \partial v^i}, \qquad H_{ij} \equiv \frac{\partial^2 L}{\partial v^i \partial v^j}.$$

Provided that $\det[H_{ij}] \not\equiv 0$ for this system, one can express accelerations $a^i = \ddot{Q}^i(\lambda)$ as independent functions of positions and velocities. When the Hessian determinant $\det(H_{ij})$ is non-vanishing the Lagrangian is called regular, otherwise it is called singular. For a singular Lagrangian, there is an infinite number of accelerations available from which a dynamical system can choose at any stage of its motion. The regularity (or singularity) is a qualitative feature, independent of the particular coordinates in which the Lagrangian has been expressed.

Note, that the discussion just above assumes that the Lagrangian has been expressed in terms of the physical degrees of freedom only. In a more general situation, the notion of a Lagrangian regularity or singularity becomes context-dependent. The reason for this is that, in describing a dynamical system, one can use a Lagrangian involving only $s$ physical degrees of freedom or a Lagrangian in an extended configuration space involving additional $r$, non-dynamical degrees of freedom. The Hessian square matrix has dimension $s$ in the first case and dimension $s + r$ in the extended case. In both cases, however, the Hessian rank must

not be lower than $s$. The use of a Lagrangian description involving spurious or auxiliary degrees of freedom often makes the description more transparent or easier to tackle with, for example, fully covariant. In order not to come into confusion, instead of referring to singularity/regularity of a Lagrangian it would be better to refer to the rank of the Hessian matrix (denoted with $\mathrm{Rk}(H)$) as it is not changed when additional gauge degrees of freedom are introduced and, accordingly, the Lagrangian is rewritten in an extended configuration space.

A good example is provided by the ordinary point particle. Its Lagrangian in the covariant form $L = -m\sqrt{\dot{x}\dot{x}}$ is singular – it involves a spurious gauge degree of freedom. In the gauge $\lambda = x^0$ one gets a regular Lagrangian $L = -m\sqrt{1-\dot{x}\dot{x}}$, where $\dot{x}$ is the spatial velocity vector. In both cases, the Hessian rank is 3 and equals the number of degrees of freedom considered physical in the context of a point particle. Similarly, when it comes to a relativistic rotator, the Lagrangian recast in a form involving only the 5 physical degrees of freedom characteristic of a genuine rotator should be regular, which means that the determinant of the corresponding 5×5 Hessian matrix must be non-vanishing. This implies that the rank of the full 8×8 Hessian matrix of the original singular Lagrangian (1) involving also non-dynamical degrees of freedom should be 5 too.

One can verify the condition $\mathrm{Rk}(H) = 5$ for all members of the considered family of relativistic rotators (1) regarded as dynamical systems with 5 physical degrees of freedom. Following the calculation presented in [5], one can start with Cartesian coordinates $(x, y, z)$ and spherical angles $(\varphi, \theta)$ describing the position and the null direction in a reference system of some inertial observer. The arbitrary parameter $\lambda$ can be set to be proportional to the time of that observer, $\lambda = l^{-1}t$. Then, in terms of the vector matrices $V = [\dot{x}, \dot{y}, \dot{z}]^T$, $N = [\sin\theta\cos\varphi, \sin\theta\sin\varphi, \cos\theta]^T$ and $W = [\dot{\theta}, \dot{\varphi}\sin\theta]^T$, the Lagrangian form (1) gets reduced to

$$L = -m\sqrt{1-V^TV}\,f(\xi), \quad \text{with} \quad \xi = \frac{W^TW}{(1-N^TV)^2} \quad \text{and} \quad f'(\xi) \not\equiv 0. \tag{3}$$

The Hessian determinant can be found by taking components of vectors $V$ and $W$ as independent velocity variables (linearly related to the original set of velocities) and using some identities for determinants of block matrices. As shown in [5], the resulting determinant reads

$$\det[H_{ij}] \propto f^3(\xi)\left[f'(\xi)\right]^2\left(1 + 2\xi\left(\frac{f'(\xi)}{f(\xi)} + \frac{f''(\xi)}{f'(\xi)}\right)\right),$$

where the proportionality factor (not shown) is independent of $f$. Hence, only with $f$ satisfying the differential equation $\left(f(\xi) + 2\xi f'(\xi)\right)f'(\xi) + 2\xi f(\xi)f''(\xi) = 0$ the Lagrangian (3) will be singular. This equation has only one solution such that $f'(\xi) \not\equiv 0$, namely

$$f(\xi) = a\sqrt{1 \pm b\sqrt{\xi}},$$

with $a$ and $b$ being positive integration constants to be set by the Casimir parameters.

Now it becomes clear that the only Lagrangian with deficient rank in the investigated family of relativistic rotators (1) is that defined by the action (2) (its Hessian rank is 4, not 5). In consequence of this the phase of the clocking mechanism has the nature of a gauge variable [5, 9], which is the reason why the dynamical system (2) cannot be interpreted as a clock.

## 4 Singularities in the inverse Legendre transformation. Zitterbewegung with the speed of light.

According to Dirac's method [10], the Hamiltonian for a (reparametrization invariant) relativistic system is a linear combination of first-class constraints (whose Poisson bracket with

all other constraints is vanishing). The coefficients of this combination are arbitrary functions of the independent parameter. There are four such constraints for the Lagrangian (2): the first two follow from the requirement imposed on both Casimir invariants: $C_1 \equiv pp \simeq m^2$ and $C_2 \equiv -\det \mathrm{Gram}(p,k,\pi) \simeq -\frac{1}{4} m^4 l^2$; the other two constraints concern the particular realisation of the spinning degrees of freedom described by a null direction $k$ (with the corresponding conjugate momentum $\pi$): $kk \simeq 0$ and $k\pi \simeq 0$ – the latter ensures that the physical state is independent of the arbitrary scale of $k$. All of these constraints are first-class. Remembering that one can use any equivalent combination of constraints, it immediately follows that the total Hamiltonian, as implied by the original Lagrangian form (2), can be taken as [9]

$$\mathcal{H} = \frac{u_1}{2m}\left[pp - m^2\right] + \frac{u_2}{2m}\left[pp + \frac{4}{l^2 m^2}(kp)^2 \pi\pi\right] + u_3 k\pi + u_4 kk\,, \qquad (4)$$

with $u_i$'s being independent arbitrary functions.[4] Now the Hamiltonian constraints follow from the equations $\partial_{u_i}\mathcal{H} = 0$ while the velocities are defined through the Hamiltonian equations:

$$\dot{x}^\mu = \frac{\partial \mathcal{H}}{\partial p_\mu} = \frac{u_1 + u_2}{m} p^\mu + u_2 \frac{4(kp)(\pi\pi)}{l^2 m^3} k^\mu\,, \qquad \dot{k}^\mu = \frac{\partial \mathcal{H}}{\partial \pi_\mu} = u_2 \frac{4(kp)^2}{l^2 m^3}\pi^\mu + u_3 k^\mu\,. \qquad (5)$$

Now, the Hamiltonian form (4) can be assumed as a starting point. All Lagrangians corresponding to the Hamiltonian (4) can be obtained by applying the inverse Legendre transformation. The form of the resulting Lagrangian $L \equiv p\dot{x} + \pi\dot{k} - \mathcal{H}$, when expressed in terms of the velocities, is subject to the invertibility of the map (5) restricted to the submanifold defined by the Hamiltonian constraints. On this submanifold induced is a corresponding map between two sets of scalar variables $\{u_1, u_2, u_3, kp, p\pi\}$ and $\{\dot{k}\dot{k}, \dot{k}\dot{x}, \dot{x}\dot{x}, k\dot{x}, k\dot{k}\}$ which is easier to investigate:

$$\dot{x}\dot{x} = u_1^2 - u_2^2\,, \qquad k\dot{x} = (u_1 + u_2)\frac{kp}{m}\,, \qquad \dot{k}\dot{k} = -\frac{4(kp)^2}{l^2 m^2}u_2^2\,,$$

$$\dot{k}\dot{x} = (u_1 + u_2)\left[\frac{4(kp)(p\pi)}{m^3 l^2}u_2 + u_3\right]\frac{kp}{m}\,, \qquad k\dot{k} = 0\,. \qquad (6)$$

The number of new constraints for velocities depends on the rank of the Jacobi matrix of the above mapping. It can be shown that this rank depends only on the variables $u_1$, $u_2$, and equals 4 for $u_1^2 \neq u_2^2 \neq 0$, 3 for $u_1 = u_2 \neq 0$, and 2 for $u_1 = -u_2 \neq 0$.

In passing from the Hamiltonian to the Lagrangian, one may first assume that $u_1 + u_2 \neq 0$ and $u_2 \neq 0$. Then the momenta expressed as functions of velocities and $u_i$'s read

$$p^\mu = \frac{m}{u_1 + u_2}\dot{x}^\mu - \frac{l^2 m (u_1 + u_2)^2 (\dot{k}\dot{k} - 2u_3 k\dot{k})}{4(k\dot{x})^2 u_2}\frac{k^\mu}{k\dot{x}}\,, \qquad \pi^\mu = \frac{l^2 m (u_1 + u_2)^2}{4(k\dot{x})^2 u_2}(\dot{k}^\mu - u_3 k^\mu)\,.$$

From the constraint equations $pp - m^2 = 0$ and $pp + \frac{4}{l^2 m^2}(kp)^2(\pi\pi) = 0$ two conditions for $u_1$ and $u_2$ follow:

$$\frac{\dot{x}\dot{x}}{(u_1 + u_2)^2} + \frac{u_1 + u_2}{2u_2}\xi = 1\,, \quad \text{and} \quad \frac{(u_1 + u_2)^2}{4u_2^2}\xi = 1\,. \qquad (7)$$

The resulting $u_1$, $u_2$ can be expressed as independent functions of the velocities, provided that the Jacobian determinant of the transformation (7) — regarded as one leading from variables

---

[4]The Hamiltonian formulation of the whole class of relativistic rotators defined by the general Lagrangian (1) was presented in [9]. This formulation uses the minimal phase space in terms of four-vectors. There is also possible a description of dynamical systems in extended phase spaces that upon reduction should recover the minimal Hamiltonian form. In the case of the particular Lagrangian (2) such an approach was presented by Das and Ghosh [11] who also obtained the Hamiltonian (4). They started with a counterpart of Lagrangian (2) written in an extended space exploiting a trick, introduced by Lukierski Stichel and Zakrzewski [12], in which additional auxiliary variables allow one to make the time derivative structure of the original Lagrangian easier to tackle with.

$(\dot{x}\dot{x}, \xi)$ to variables $(u_1, u_2)$ which, up to a constant factor, is equal to $\frac{\xi \dot{x}\dot{x}}{u_2^3(u_1+u_2)}$ — is non-zero. In this case the resulting Lagrangian overlaps with that in the action integral (2). However, assuming that the condition $\dot{x}\dot{x} \neq 0$ is not satisfied, two other Lagrangians are possible.

In the first case $u_1 = u_2$, and the corresponding new velocity constraints follow:

$$\frac{\dot{x}\dot{x}}{k\dot{x}} = 0, \qquad l^2 \frac{\dot{k}\dot{k}}{k\dot{x}} + k\dot{x} = 0.$$

Then, from (6), $u_1 = \chi$, $u_2 = \chi$, $u_3 = \nu$, $kp = \frac{m}{2\chi}k\dot{x}$ and $p\pi = \frac{l^2m^2}{2k\dot{x}}\left[\frac{k\dot{x}}{k\dot{x}} - \nu\right]$ with $\chi$ and $\nu$ being arbitrary functions. After discarding a total derivative involving $\dot{k}\dot{k}$ and the higher order terms in the velocity constraints, the resulting Lagrangian can be cast in the following form linear in these constraints

$$L = \frac{m\kappa}{2}\frac{\dot{x}\dot{x}}{k\dot{x}} + \frac{m}{4\kappa}\left[l^2 \frac{\dot{k}\dot{k}}{k\dot{x}} + k\dot{x}\right] + \Lambda \, kk. \tag{8}$$

Here, $\kappa(\chi) \equiv \frac{kp}{m}$ is a new variable independent of velocities while $\Lambda$ is a Lagrange multiplier.

In the second case, for $u_1 = -u_2$, a restricted Legendre transformation should be considered with $p^\mu$ left (for a while) unaltered. Using equations (5) and (6), one can find that $\pi = \mp\frac{lm^2}{2}\frac{\dot{k}-u_3 k}{kp\sqrt{-\dot{k}\dot{k}}}$ and $u_2 = \mp\frac{lm}{2kp}\sqrt{-\dot{k}\dot{k}}$. Now, integrating away the term linear in $\dot{k}\dot{k}$, another Lagrangian is obtained in the form

$$L = p\dot{x} \pm \frac{lm^2}{2}\frac{\sqrt{-\dot{k}\dot{k}}}{kp} + \Lambda \, kk. \tag{9}$$

Inferred from equations (5) and (6) the result $\dot{x}^\mu = \pm\frac{lm^2}{2}\frac{\sqrt{-\dot{k}\dot{k}}}{(kp)^2}k^\mu$ can be re-obtained by performing arbitrary variations of the Lagrangian with respect to $p^\mu$, hence $e\dot{x} = \pm\frac{lm^2}{2}\frac{\sqrt{-\dot{k}\dot{k}}}{2(kp)^2}ek$ for any vector $e^\mu$, and this fact can be used to eliminate $p^\mu$ from (9). Accordingly, the alternative form of the above Lagrangian can be taken to be

$$L = m\left[\frac{-4l^2\dot{k}\dot{k}}{(ek)^2(e\dot{x})^2}\right]^{1/4}e\dot{x} + \Lambda \, kk,$$

which involves arbitrary (timelike) $e^\mu$ (then the condition $ek \neq 0$ is satisfied) playing the role of the initial momentum $p$.

Unlike the Lagrangian (2), the new Lagrangians (8) and (9) have analytic structure compatible with the constraint $\dot{x}\dot{x} = 0$. They describe intrinsic motion with the speed of light (see Appendix).

## 5 Conclusion

In this paper, the present status of Staruszkiewicz's relativistic rotators in free motion was discussed. The original motivation behind introducing the rotators was the idea of devising a model of an ideal clock that could be used to test the clock hypothesis [4]. However, the constraints imposed on the Casimir invariants for the purpose of realising the quantum irreducibility idea on the classical level, lead to Lagrangians with deficient Hessian rank (which is 4 instead of 5) when subluminal intrinsic motion is assumed from the start. In consequence of this the clocking rate remains arbitrary function of the proper time in the momentum rest frame.

However, at the level of constrained Hamiltonians, one makes no a priori assumptions about the velocities. Constraints on velocities may appear when passing from the Hamiltonian to the Lagrangian. With this method one recovers the original Lagrangian with subluminal motion when the rank of the inverse Legendre transformation is maximal. For a lower rank (when this transformation becomes singular) one obtains two new Lagrangians (8) and (9) with intrinsic motion with the speed of light (the motion of the momentum rest frame is still subluminal). The solutions are presented in Appendix.

The dynamical systems described by the new Lagrangians exhibit behaviour that can viewed as a counterpart of Zitterbewegung known for two states of Dirac's free electron (see the interesting and original discussion by Breit [13]). The existence of the two systems conforms with the distinguished role of the constraint $\dot{x}\dot{x} = 0$. It remains to investigate how these systems would behave when appropriately coupled with the electromagnetic or gravitational field.

## A   Appendix

For both Lagrangians, the momentum $P \equiv \partial_{\dot{x}}$ is conserved, hence $P = me$, where $e$ is a constant unit future-oriented timelike four-vector and $m$ is a mass parameter. The equation $\partial_\Lambda L = 0$ implies $kk = 0$. The arbitrary parameterization $\lambda$ and the arbitrary scale of $k$ can be chosen so that $e\dot{x} = 1$ and $ke = 1$. Furthermore, the spatial vector $n$ defined by $k = e + n$ is unit and orthogonal to $e$: $nn = -1$ and $ne = 0$.

For the Lagrangian (9), the equation $\partial_p L = 0$ implies $\dot{x} = (l/2)\Omega(e+n)$ with $\Omega \equiv \sqrt{-\dot{n}\dot{n}}$, which in turn gives $\Omega = 2/l$ from the previous condition $e\dot{x} = 1$ which is now seen to identify $\lambda$ with the time $t$ in the momentum frame (in which the time axis is directed along $e$). Finally, $\dot{x} = e + n$. The momentum $\Pi \equiv \partial_{\dot{k}} L$ reduces to $\Pi = -(ml^2/4)\dot{n}$. Since $\partial_k L = -me + 2\Lambda(e+n)$, the respective Lagrangian equation reduces to the equation for large circles on a unit sphere, $\ddot{n} + (2/l)^2 n = 0$, where the Lagrange multiplier $\Lambda = m/2$ was earlier determined upon taking the scalar product $n(\dot{\Pi} - \partial_k L) = 0$ and using the identity $\dot{n}\dot{n} + n\ddot{n} = 0$ satisfied by any vector with constant product $nn$. The solution reads $n = a\cos\phi + b\sin\phi$, where $\phi = (2/l)t$ is the phase, $a$ and $b$ are constant vectors such that $aa = -1 = bb$, $ab = 0$, $ae = 0 = be$. Substituting this in the other equation for $x$ and integrating, one obtains $x = et + (l/2)(a\sin\phi - b\cos\phi)$. The phase $\phi = (2/l)t$ is a unique function of the proper time $t$ in the momentum frame and $\dot{x}\dot{x} = 0$.

For the Lagrangian (8), the conserved momentum $P \equiv \partial_{\dot{x}} L = me$ implies

$$\frac{\kappa}{k\dot{x}}\dot{x} = e - \frac{1}{4\kappa}\left(1 - \frac{l^2\dot{k}\dot{k} + 2\kappa^2\dot{x}\dot{x}}{(k\dot{x})^2}\right)k, \quad \text{hence} \quad \kappa = ek.$$

Now, taking scalar products of the above equation with $e$ and $\dot{x}$, and with itself, one gets three equations from which one finds that $k\dot{x} = 2(ke)(e\dot{x})$, $l^2\dot{k}\dot{k} + (k\dot{x})^2 = 0$, and $\dot{x}\dot{x} = 0$. This in turn implies $\dot{x}/(e\dot{x}) = 2e - k/(ke)$. By applying the gauge $e\dot{x} = 1$ and $ke = 1$ as in the previous case, and then the decomposition $k = e + n$, one finally obtains $\dot{n}\dot{n} = -4/l^2$ and $\dot{x} = e - n$ (note the sign difference with the previous case). Then one finds in an analogous way as before, that $\ddot{n} + (2/l)^2 n = 0$, however with $\Lambda = -m/4$. This leads to a solution $x = et - (l/2)(a\sin\phi - b\cos\phi)$ with $\phi = (2/l)t$.

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
