# Peer review of "Clocking mechanism from a minimal spinning particle model"

_SciPost Physics Proceedings, doi:SciPost Phys. Proc. 14, 042 (2023)_

## Round 1 · Referee Report · Etera R. Livine (Referee 1) · 2023-1-10

Report
Although announcing (in the abstract and introduction) a study of clocks in relativity, this manuscript actually proposes short analysis of the standard massive spinning particle. Besides vague speculations in the introduction and conclusion, the goal appears to be to study the Lagrangian and Hamiltonian formulation of a particle carrying a fundamental representation of the Poincaré group, i.e. fixed mass and spin. This is standard mechanics, it is hardly original and the presentation by the authors show a misconception of the problem. Indeed, what they call a singular Lagrangian is more simply symptomatic of a gauge symmetry, coming from first class constraints, as known and expected. If one were to follow the logic developed by the atuhors (more precisely, section 3), one would discard the standard (geodesic) Lagrangian for the (massive spinless) relativistic particle. In fact, it is known -standard text book physics- how to deal with such a Lagrangian. The Hamiltonian is a linear combination of the constraints, the coefficients in front of those constraints are Lagrange multipliers and can be chosen arbitrarily (with, of course, appropriate smoothness and monotonicity assumptions). Such a choice here of u1,u2,u3 and u4 of eqn (3)) amounts to a gauge fixing of the constraints, this yields Lagrangian whose equations of motion are well-defined and describe the evolution of the gauge-fixed system prescribed by the choice of Lagrange multipliers.
Due to the weakness and non-originality of the analysis, I can not recommend this manuscript for publication. If the authors persevere in this line of research, I would urge them to revise their work in light of the well-known Hamiltonian formulation of constrained systems, and gauge symmetries, and to solve the evolution equations explicitly to actually discuss clock properties.
Due to the weakness and non-originality of the analysis, I can not recommend this manuscript for publication. If the authors persevere in this line of research, I would urge them to revise their work in light of the well-known Hamiltonian formulation of constrained systems, and gauge symmetries, and to solve the evolution equations explicitly to actually discuss clock properties.
Author: Tobiasz Pietrzak on 2023-01-19 [id 3250]
(in reply to Report 1 by Etera R. Livine on 2023-01-10)Due to the elaborate answer containing mathematical expressions, it has been attached as a PDF file.
Attachment:
Answer.pdf

---

## Editorial Decision

published